# *FvMYB108*, a MYB Gene from *Fragaria vesca*, Positively Regulates Cold and Salt Tolerance of *Arabidopsis*

**DOI:** 10.3390/ijms25063405

**Published:** 2024-03-17

**Authors:** Penghui Song, Ruihua Yang, Kuibao Jiao, Baitao Guo, Lei Zhang, Yuze Li, Kun Zhang, Shuang Zhou, Xinjuan Wu, Xingguo Li

**Affiliations:** 1Research Institute of Rural Revitalization Technology, Heilongjiang Academy of Agricultural Sciences, Harbin 150028, China; ssph2996@163.com (P.S.); zxsgss@haas.cn (K.J.); btg1974@haas.cn (B.G.); zhang-lei18@163.com (L.Z.); liyz2023@163.com (Y.L.); wysll21@163.com (K.Z.); fengzhongxu@163.com (S.Z.); xiaojuan414@163.com (X.W.); 2Horticulture Branch of Heilongjiang Academy of Agricultural Sciences, Harbin 150040, China; yrh_ray@126.com; 3Key Laboratory of Biology and Genetic Improvement of Horticultural Crops (Northeast Region), Ministry of Agriculture and Rural Affairs, National-Local Joint Engineering Research Center for Development and Utilization of Small Fruits in Cold Regions, College of Horticulture & Landscape Architecture, Northeast Agricultural University, Harbin 150030, China

**Keywords:** abiotic stress, gene cloning, overexpression, regulatory mechanism

## Abstract

MYB (myoblast) protein comes in large quantities and a wide variety of types and plays a role in most eukaryotes in the form of transcription factors (TFs). One of its important functions is to regulate plant responses to various stresses. However, the role of MYB TFs in regulating stress tolerance in strawberries is not yet well understood. Therefore, in order to investigate the response of MYB family members to abiotic stress in strawberries, a new MYB TF gene was cloned from *Fragaria vesca* (a diploid strawberry) and named *FvMYB108* based on its structural characteristics and evolutionary relationships. After a bioinformatics analysis, it was determined that the gene belongs to the R2R3-MYB subfamily, and its conserved domain, phylogenetic relationships, predicted protein structure and physicochemical properties, subcellular localization, etc. were analyzed. After qPCR analysis of the expression level of *FvMYB108* in organs, such as the roots, stems, and leaves of strawberries, it was found that this gene is more easily expressed in young leaves and roots. After multiple stress treatments, it was found that the target gene in young leaves and roots is more sensitive to low temperatures and salt stimulation. After these two stress treatments, various physiological and biochemical indicators related to stress in transgenic *Arabidopsis* showed corresponding changes, indicating that *FvMYB108* may be involved in regulating the plant’s ability to cope with cold and high-salt stress. Further research has found that the overexpression of this gene can upregulate the expression of *AtCBF1*, *AtCOR47*, *AtERD10*, and *AtDREB1A* related to low-temperature stress, as well as *AtCCA1*, *AtRD29a*, *AtP5CS1*, and *AtSnRK2.4* related to salt stress, enhancing the ability of overexpressed plants to cope with stress.

## 1. Introduction

The strawberry (*Fragaria × ananassa Duch.*), known as the “Queen of Fruits”, is rich in various nutrients and compounds. Eating strawberries is beneficial for physical health [1,2]. Moreover, strawberries are an important economic fruit tree, which help fruit farmers increase income, improve agricultural structure, and have a wide planting range. However, unfavorable environmental factors such as cold and saline–alkaline can threaten the normal growth and development, yield, and fruit quality of strawberries [3]. The cultivation and management of strawberries require precise and strict control, because their roots are shallow and highly sensitive to osmotic stress, and they have weak salt tolerance [4]. Similarly, strawberries are also highly susceptible to the influence of low temperatures, so they require good growing conditions. However, because the overall lower level of facilities in China compared to foreign countries, cold has become one of the main limiting factors for the development of strawberries [5]. In order to reduce the damage caused by abiotic stress to the strawberry industry, in addition to improving environmental conditions, it is more effective to explore key genes through genetic engineering breeding and enhance the adaptability of fruit trees to adversity [6].

Numerous studies have demonstrated the presence of different transcription factors (TFs) in plants, which bind to gene promoters to regulate the transcription of target genes [7]. The MYB protein is named after the N-terminally conserved MYB domain and has a large number of diverse functions in plants. In plants, MYB proteins exhibit diverse functions, participating in the regulation of physiological activities, secondary metabolism, and responses to abnormal growth conditions through molecular mechanisms [8,9,10]. The typical feature of MYB TFs is that their conserved domain is MYB-DNA, which contains 1–4 incomplete repetitive sequences (R). Based on the number of Rs in the domain, the family is divided into four subclasses: 1R-MYB, R2R3-MYB, 3R-MYB, and 4R-MYB [11,12,13]. Among these, R2R3-MYB has the richest quantity and highest proportion in plants and has complex functions. It can not only play a role in the synthesis of various secondary metabolites, but also participate in plant response to various environmental factors that are not conducive to survival and hormone responses [14,15], thus holding immense significance for plant growth and development.

The mechanism by which MYB transcription factors enhance plant adaptability to environmental stress is intricate, requiring the coordinated action of multiple signal transduction pathways to elicit responses that promote robust growth under adverse conditions. Studies have shown that the R2R3-MYB TF can bolster plant stress tolerance by modulating the expression of stress-related genes through the ABA signaling pathways [16]. *AtMYB96* enhances a plant’s survival ability in low temperatures and arid environments through the ABA signaling pathway [17,18]. Overexpression of *SiMYB19* in foxtail millet (*Setaria italica* (L.) *P*. Beauv.) can upregulate the expression of *OsNCED3* related to ABA synthesis, as well as *OsPK1-* and *OsABF2*-related genes in ABA-dependent pathways, thereby reducing the sensitivity of transgenic foxtail millet to high salt. The SOS signaling pathway is also an important pathway for plants to respond to stress. The SOS1 protein is involved in the reverse transport of Na^+^/H^+^, allowing plant cells to maintain normal life forms and activities under high salt concentrations [19,20]. The CBF-dependent pathway is a key pathway for plants to respond to low temperatures [21], through which R2R3-MYB TF can play a role. The CBF promoter has a recognition sequence for this subclass of transcription factors, MYB15, which activates downstream MYB genes, enabling plants to cope with cold stimuli [22]. For example, *MdMYB88/124* can stimulate *MdCA1* and *MdCOR47* through signal transduction pathways upon receiving low-temperature signals, and *MdCBF1/MdCBF3* and other related genes were highly expressed [23]. Low temperatures can induce the expression of the *IlMYB306* gene in *Iris laevigata*, and the content and activity of various physiologically active substances in tobacco overexpressing *IlMYB306* undergo significant changes after low-temperature stress, such as a decrease in chlorophyll content, an increase in free proline content, and a significant increase in activities such as MDA, SOD, and POD [24].

In this study, a new MYB gene was isolated from *F. vesca* using gene-cloning technology and named *FvMYB108*. The evolutionary relationship, protein structure, subcellular localization, expression in various tissues, and response to stress were analyzed. Based on these results and previous research, it is speculated that this gene may play a role in the response of plants to low temperatures and drought stress. To verify this hypothesis, this gene was transferred to Arabidopsis to characterize its function in plant salt and cold tolerance. The research results confirm our hypothesis that *FvMYB108* can enhance plant tolerance to low temperatures and salt stress by regulating genes related to these two stresses. This experiment lays the foundation for studying the function of R2R3-MYB-TFs and provides a theoretical basis and candidate genes for strawberry breeding.

## 2. Results

### 2.1. Cloning and Bioinformatics Analysis of FvMYB108

RNA was extracted from *F. vesca* leaves and reverse-transcribed into cDNA as an amplification template. The product was sequenced, and the sequencing results show that *FvMYB108* had a total length of 1023 bp (Appendix A). Then, the predicted protein was analyzed, and it was found that its molecular weight was 38.15 kDa, containing 340 amino acids. Among the 20 amino acids, the four highest proportions were Ser (10.6%), Asn (9.4%), Ala (7.6%), and Leu (7.4%). The isoelectric point of this protein was 5.99, indicating hydrophilicity (grand average of hydrophilicity = −0.790).

After a BLAST analysis of the target gene sequence, nine MYB proteins with high similarity were selected; they were *MdMYB108*-like, *MsMYB108*, *PaMYB108*, *PbMYB108*-like, *PeMYB108*-like, *RcMYB08*, *RhMYB108*, and *RrMYB108*. After comparing the sequences of these proteins, it was found that the FvMYB108 protein, like these proteins, contains a special structural domain of the R2R3-MYB subfamily, namely a conserved R2R3 domain. The alignment results are shown in Figure 1A. In MEGA7.0 software, a phylogenetic tree was constructed, and the results are shown in Figure 1B. It can be seen that the closest phylogenetic relationship to FvMYB108 was PeMYB108-like, indicating a similar biological function between the two.

### 2.2. Structural Prediction of FvMYB108 Protein

The secondary structure analysis of the FvMYB108 protein is shown in Figure 2A, which shows that the alpha helix and random coil have the highest content. After a SMART program analysis, it was found that its domain contained two SANT domains, which are consistent with the characteristics of the R2R3-MYB subfamily and belong to this family (Figure 2B). The tertiary structure of the protein is shown in Figure 2C, which was consistent with the predicted results, further indicating that the secondary protein was a member of the R2R3-MYB subfamily.

### 2.3. Subcellular Localization of FvMYB108

We injected Agrobacterium tumefaciens solution containing 35S: FvMYB108-GFP and 35S:: GFP vectors into tobacco leaves and performed transient expression in their epidermal cells. Observing the expression under confocal microscopy, it can be seen that the fluorescence of the 35S:: GFP construct as a control can be observed throughout the entire cell. However, unlike the control group, the fluorescence of the 35S: FvMYB108-GFP fusion protein can only be seen in the nucleus (Figure 3E). Based on this, it can be concluded that the protein is a nuclear protein and only exists in the nucleus.

### 2.4. Expression of FvMYB114 in Strawberry Tissue

The expression level of *FvMYB108* varies in different tissues and has tissue specificity (Figure 4A). It was more easily expressed in young leaves and roots, while the expression level was lower in mature leaves and stems. As a control, the expression levels in mature leaves were 1.06 times, 1.90 times, and 2.31 times higher in the stems, roots, and young leaves, respectively. Taking the young leaves and roots as the next research objects, it can be found that the expression level of *FvMYB108* reached its highest value at different times after being subjected to cold, salt, drought, heat, and ABA stress treatments. The time to reach the highest expression level under the five treatments in young leaves was 2 h, 4 h, 6 h, 2 h, and 6 h, respectively (Figure 4B), while in roots, it was 4 h, 4 h, 6 h, 6 h, and 2 h, respectively (Figure 4C). In both tissues, *FvMYB108* was able to respond faster under cold and high-salt stimuli and still showed high expression levels after 12 h of treatment, indicating that *FvMYB108* was a gene that can be induced by low temperatures and salt.

### 2.5. Overexpression of FvMYB108 Enhances Cold Tolerance in Arabidopsis

The control group’s WT lines and UL lines (the vectors transferred in these lines are not connected to the target gene), as well as the three lines with the highest expression levels were selected based on the qPCR detection results, namely L1, L4, and L5 (Figure 5A), and were placed together in a low-temperature environment at 4 °C for growth. After 14 h, their phenotypic changes were observed. According to the data presented in Figure 6B, it is evident that the WT and UL lines exhibited more pronounced wilting and leaf damage after exposure to cold stress. Additionally, there was a noticeable yellowing trend in leaf color in comparison to the transgenic lines. Subsequently, these plants were placed back in the incubator and after 7 d of recovery, it was found that the control group had significantly withered, yellowed, and even turned white. The calculated survival rates were 23.4% and 24.5%, while most transgenic lines maintained normal growth, with survival rates of 75.2%, 80.11%, and 78.36% (Figure 5C).

In order to further test whether transgenic *Arabidopsis* undergoes corresponding changes to adapt to cold environments after being subjected to low-temperature stress, several major physiological indicators related to plant stress resistance were measured. It was found that there was no significant difference in these indicators under normal conditions. However, after 14 d of low-temperature treatment, the chlorophyll content decreased, but the changes varied among the different lines. The decrease in chlorophyll content in L1, L4, and L5 was less than that in the WT and UL systems. Although the content of MDA and proline increased, the degree of increase was different. For MDA, the increase was more in the WT and UL lines, while for proline, the increase was more in the transgenic lines. In addition, CAT, SOD, and POD activities were also enhanced, and a higher activity was observed in L1, L4, and L5. These data (Figure 6) indicate that overexpression of *FvMYB108* can enable plants to adapt to environmental changes in cold environments, rapidly clearing reactive oxygen species (ROS) and enhancing their adaptability to this unfavorable environment for survival.

### 2.6. Overexpression of FvMYB108 Regulates the Expression of Cold Tolerance-Related Genes

TFs can regulate the expression of target genes by binding to their promoters, which plays a crucial role in plant stress tolerance. In this study, qPCR detection was performed on the expression levels of four genes related to cold and salt stress in *A. thaliana* overexpressing *FvMYB108*. The detection results are shown in Figure 7, and it can be seen that under suitable growth conditions, the expression levels of these four genes are not significantly different, and their expression levels are all very low. However, after growing in a cold environment, the expression levels change. Although they all increased, the expression levels were higher in L1, L4, and L5. These results indicate that *FvMYB108* can endow plants with stronger survival ability in cold environments by regulating the expression of *AtCBF1*, *AtCOR47*, *AtERD10*, and *AtDREB1A*.

### 2.7. Overexpression of FvMYB108 Enhances Salt Tolerance in Arabidopsis

After being subjected to salt stress for 7 d, the phenotypic changes in WT, UL, L1, L4, and L5 plants were observed and recorded. After being subjected to salt stress, WT and UL plants exhibited severe wrinkling and shrinkage, and a large area of chlorosis, yellowing, and even browning occurred (Figure 8A); In contrast, the overexpression of *FvMYB108* in *Arabidopsis* was less affected and the degree of change was not significant, resulting in better survival rates of 74.61%, 78.51%, and 71.44%, respectively. However, WT and UL strains only had survival rates of 28.45% and 31.32%, as shown in Figure 8B.

After salt stress treatment, relevant physiological indicators were measured, and the results are shown in Figure 9. It can be seen that before treatment, the content of chlorophyll, MDA, and proline, as well as the activities of CAT, SOD, and POD, were very low, and the levels were similar in all plant lines. However, after salt stress, the chlorophyll content decreased, especially in the WT and UL lines, and the content of MDA and proline increased to varying degrees, and the MDA content was higher in the WT and UL lines. While the proline content was the opposite, its content was higher in the L1, L4, and L5 lines. CAT, SOD, and POD activities were also significantly increased, and the activity in the transgenic lines was significantly higher than that in the WT and UL lines. These results provide evidence that the overexpression of *FvMYB108* can improve plant adaptation to high-salt environments and the ability to clear ROS after stress.

### 2.8. Overexpression of FvMYB108 Regulates the Expression of Salt Tolerance-Related Genes

In this study, qPCR detection was performed on the expression levels of four known salt stress-related genes in *FvMYB108* overexpressing *A. thaliana*. The results show that in Figure 10, under normal growth conditions, the expression levels of these four genes were very low and almost indistinguishable. However, after high salt concentration treatment, the expression levels of the genes increased. However, there is a significant difference in the expression levels between the WT and UL lines and the L1, L4, and L5 lines, with higher expression levels in the transgenic lines. This indicated that *FvMYB108* can better adapt to high-salt environments by positively regulating the expression of *AtCCA1*, *AtRD29a*, *AtP5CS1*, and *AtSnRK2.4*.

## 3. Discussion

Low temperatures and soil salinization can pose a threat to the production and quality of strawberries, affecting the development of the strawberry industry [25,26]. Faced with various threats during growth and development, plants have evolved complex regulatory mechanisms to adapt to various adverse environments that may occur. The regulation of related genes by TFs is the key to these mechanisms; therefore, exploring candidate genes for cold and salt tolerance in strawberries is a key means of strawberry breeding [27]. There are many types of TFs in eukaryotes, among which MYB TFs are the most widely distributed and have complex functions, including metabolic reactions, cell morphological changes, development and other life activities, as well as stress responses to various extreme environments [12,28]. However, research on the role of the strawberry MYB TF family in plant response to stress is still limited, and there are currently few reports. To fill this gap, we took strawberries as the research object, extracted RNA from their leaves, and used gene-cloning technology to obtain a new MYB gene. A bioinformatics analysis of this gene can reveal its related properties, and after analyzing its structure, it was found that it belongs to the R2R3-MYB subfamily. This provides a basis for studying its function. After further analysis of its genetic relationship, it was found that PeMYB108-like has the highest homology with the target protein. Therefore, based on the function of PeMYB108-like, it can be inferred that FvMYB108 may play a role in plant response to abiotic stress, laying the foundation for further research and providing ideas.

In numerous plant species, it has been demonstrated that members of the R2R3-MYB subfamily possess the ability to confer stress tolerance by regulating the expression of stress-related genes in response to unfavorable environmental conditions. For instance, *CsMYB30* cloned from sweet orange can participate in the process of transgenic plants responding to salt and drought stress [29]. After being transformed into apple callus and *Arabidopsis*, *MdMYB23* directly binds to the promoter of *MdCBF1/2* and upregulates these two genes, thereby reducing the cold injury point in plants [30]. Additionally, the activation of the peroxidase gene by AtMYB49 enhances plant salt tolerance through the ABA signaling pathway. Our research results are also consistent with previous studies showing that *FvMYB108* has a tissue-specific expression in strawberry varieties, was more easily expressed in young leaves and roots, and was more sensitive to cold and salt stimuli. To further verify the function of this gene in plant response to cold and salt stress, it was transferred into *Arabidopsis* through genetic transformation. Based on the qPCR detection results, three transgenic lines with higher expression levels were screened, namely L1, L4, and L5. After observing and recording the phenotypic changes in each line before and after stress, it was found that after growing in low-temperature and high-salt environments for a period of time, both the WT and UL lines exhibited severe wilting and chlorosis, and their survival rates were extremely low. However, most transgenic lines were able to maintain their normal morphology, and their survival rates were above 70%, indicating that the overexpression of *FvMYB108* played a role in the plant’s cold and salt resistance.

After being exposed to environmental stress, plants undergo various physiological and biochemical changes. When receiving cold and salt signals from the outside, the enzyme system and the activity of photosynthetic pigments in photosynthesis are affected, causing changes in the composition and content of photosynthetic pigments. Notably, the chlorophyll content decreases, resulting in the yellowing and wilting of plants [31,32]. Under environmental stress, membranes are damaged by lipid peroxidation, and as the final product of this reaction, the content of MDA has become one of the important indicators to measure the degree of stress on plants [33]. The proline content also changes under stressful conditions, as external factors disrupt the osmotic balance of plant cells. An increased proline content helps stabilize cell structures and clear reactive oxygen species (ROS) [34,35]. Under normal circumstances, the concentration of ROS in cells is very low, otherwise it can cause cell oxidation and pose a threat to plants. However, when faced with adversity, the concentration of ROS will increase [36]. Faced with this situation, plants have their own coping mechanisms to clear excess ROS. SOD, CAT, and POD are key enzymes for clearing reactive oxygen species, and they work together to decompose and reduce O_2_^−^, H_2_O_2_, and OH. Therefore, when plants are affected by stress, the activity of these three enzymes increases [37,38,39]. In this experiment, it was found that after cold- and salt-stress treatment, the chlorophyll content decreased less in the transgenic lines. The MDA and proline contents, as well as the activities of CAT, SOD, and POD, all changed in a direction that is conducive to plant adaptation to adversity, allowing transgenic plants to quickly eliminate reactive oxygen species and maintain normal life activities under stress.

In the process of plants responding to adverse environmental stress, multiple genes are involved in regulation. Researchers found that in *CmICE2* transgenic Arabidopsis, the expression levels of the *CBF* and *COR* genes were significantly upregulated after low-temperature stress, indicating their responsiveness to cold stimuli [40]. Additionally, the high expression of the *MbWRKY40* gene in *Arabidopsis* promoted the accumulation of genes related to low-temperature stress, such as *AtDREB2A*, *AtRD29A*, *AtERD10*, and *AtCOR47A*, enhancing the plant’s cold tolerance [41]. Under high-salt conditions, the expression of the ABA synthesis-related gene *AtNCED3* and the ABA signaling-related gene *AtSNRK2.4*, as well as genes related to the ABA pathway like *AtCAT1*, *AtP5CS1*, and *SOS1*, was upregulated in transgenic *Arabidopsis*, associated with the SOS signal transduction pathway [31]. Furthermore, numerous studies have revealed the important role of the transcription factor MYB in plant stress responses, enhancing plant tolerance by regulating the expression of stress-responsive genes. To further understand the regulatory mechanism of *FvMYB108* in plant response to low-temperature and salt stress, we investigated the expression of several genes related to low-temperature and salt stress in overexpressing plants.

Therefore, based on the results obtained from this study and combining previous research, we have inferred a possible mode of *FvMYB108* regulating plant response to low-temperature and salt stress, as shown in Figure 11. When plants sense external cold signals, *FvMYB108* is overexpressed and then directly or indirectly upregulates the expression levels of *AtCBF1*, *AtCOR47*, *AtERD10*, and *AtDREB1A* through the CBF pathway, thereby enhancing the plant’s ability to cope with low-temperature stress. When salt stress is felt, *FvMYB108* regulates the upregulation of salt stress-related genes such as *AtCCA1*, *AtRD29a*, *AtP5CS1*, and *AtSnRK2.4* through the ABA pathway, which enhances the plant’s ability to clear reactive oxygen species and adapt to salt stress.

## 4. Materials and Methods

### 4.1. Plant Materials, Growth Conditions, and Treatment

The experimental material used in this study is *Fragaria vesca*. Prior to the study, seedlings were inoculated into nutrient soil containing vermiculite, which is half the volume of the nutrient soil. We placed plastic flower pots in a greenhouse incubator with a light cycle of 16 h and darkness of 8 h, at a temperature of 25 °C, and a relative humidity of 70% [42]. When 9–10 true leaves had grown, 30 seedlings with good growth were selected and divided evenly into 6 groups for the following treatment: the first group served as a control and grew normally in the greenhouse incubator; the second group was subjected to low-temperature stress in a −4 °C incubator; the third group contained irrigated seedlings with 200 mM NaCl for salt stress; the fourth group used 15% PEG6000 for irrigation to simulate drought stress; the fifth group was subjected to high-temperature stress in a 37 °C incubator; 100 μM ABA irrigated the sixth group of seedlings for ABA treatment. According to previous studies, samples were taken from young leaves, old leaves, roots, and stems at 0, 1, 2, 4, 8, and 12 h and frozen in liquid nitrogen before being stored at −80 °C [9,43].

### 4.2. Cloning and Vector Construction of FvMYB108

We extracted RNA from the samples according to the instructions of the OminiPlant RNA Kit (Conway Collection, Beijing, China), purified them as a template, reverse-transcribed them into cDNA, and used the TransScript first-strand cDNA synthesis SuperMix (TransGen Biotech, Beijing, China) kit. A pair of specific primers was designed, which were based on the CD region of FvMYB108, with primer sequences FvMYB108-F and FvMYB108-R displayed in Appendix A. We used the cDNA of untreated mature leaves as the amplification template to obtain the target gene, cloned the purified product onto the PEASY-T1 vector (TransGen Biotech, Beijing, China), and sent the successfully connected product to sequencing [42].

### 4.3. Subcellular Localization of FvMYB108

Based on the target gene sequence, BamHI and SalI on the pSAT6-GFP-N1 vector were selected as cleavage sites, and corresponding primers FvMYB108 silF and FvMYB108 silR were designed (Appendix A). After using restriction endonucleases for dual-enzyme digestion, a transient expression vector of FvMYB108 was constructed. Agrobacterium containing the vector was shaken and cultured to an OD600 of around 0.8. After centrifugation, the bacterial liquid was adjusted to an OD600 of around 0.2 using resuspension. The same method was used to obtain an infection solution containing GFP. The infection solution was injected into the outer skin cells of tobacco leaves using injection method. We observed the position of FvMYB108-GFP fusion protein in cells under confocal microscopy [44,45].

### 4.4. Evolutionary and Structural Analyses of FvMYB108

We went on the EMBOSS Needle website (http://www.ebi.ac.uk/Tools/psa/emboss_needle/, accessed on 15 January 2023.) to analyze the sequencing results of *FvMYB108* and translate the nucleic acid sequence into an amino acid sequence using DNAMAN5.2. Based on the target protein sequence, the BLAST function was used on the NCBI website to screen for 9 other highly homologous MYB proteins, namely MdMYB108-like (*Malus domestica*, XP_008390410.1), MsMYB108 (*Malus sylvestris*, XP_050120074.1), PaMYB108 (*Populus alba*, XP_034932063.1), PbMYB108-like (Pyrus x bretschneideri, XP_009372081.1) and PeMYB108-like (*Populus euphratica*, XP_011027944.1), RcMYB08 (*Rosa chinensis*, XP-024196866.1), RhMYB108 (*Rosa hybrid*, QEV83955.1), and RrMYB108 (*Rosa rugosa*, XP-061994110.1). Then, sequence alignment was performed in DNAMAN 5.2, and a phylogenetic tree was constructed using MEGA7 to gain a more intuitive understanding of the phylogenetic relationship of FvMYB108. The primary structure, domain, and tertiary structure of FvMYB108 protein were predicted and analyzed on ExPASy (https://web.expasy.org/protparam/, accessed on 21 January 2023), SMART (http://smart.embl-heidelberg.de/, accessed on 21 January 2023), and SWISS-MODEL (https://swissmodel.expasy.org/, accessed on 21 January 2023) websites, respectively.

### 4.5. Fluorescence Quantitative Analysis

Based on the sequence of *FvMYB108*, qPCR primers were designed on Primer Premier 5.0 to detect the expression of *FvMYB108* in various tissues under different stress and treatment times. The actin gene (XM:011471474.1, *F. vesca*) was used as an internal reference gene, and the qPCR primers were *FvMYB108*-qF/qR and *FvActin*-F/R (Appendix A). The reaction system was 2xMix 12.5 μL, cDNA 1.5 μL, each upstream and 1 downstream primer 1 μL, and we added ddH2O to make up for 25 μL. The reaction procedure followed the previous method of pre-deformation at 94 °C for 30 s, followed by denaturation at 95 °C for 5 s, annealing at 54 °C for 40 s, and annealing at 70 °C for 30 s, for a total of 40 cycles. The expression levels of target genes under different conditions were analyzed using the 2^−∆∆Ct^ method.

### 4.6. Genetic Transformation of Arabidopsis

The target fragment of FvMYB114 was fused with the PCAMBIA2300 vector through double-enzyme digestion. We transferred *A. tumefaciens* GV3101 into *Arabidopsis* Col-0 through inflorescence infection using bacterial solution containing PCAMBIA2300-FvMYB108 overexpression vector. Positive strains were screened in MS medium, and the antibiotic contained in the medium was 50 mg/L kanamycin. We performed qPCR identification on the selected plants, and gradually screened to obtain homozygous T_3_ generation transgenic plants. After analyzing the expression levels of the selected plants, the three highest expression lines were selected for further research, with WT and UL lines as controls.

### 4.7. Identification of Cold and Salt Tolerance in Arabidopsis thaliana

After disinfecting the seeds of the three selected transgenic lines, as well as the WT and UL lines, they were seeded in 1/2 MS medium. They were first vernalized at 4 °C for 3 d, then placed under light. After germination, these young seedlings were transferred to black square pots filled with nutrient soil, and one plant was planted at the four diagonal corners of each small pot. After 3 weeks of growth, these Arabidopsis plants were subjected to stress treatment, and the plants subjected to low-temperature stress were placed at −4 °C. After 14 h, they were returned to their original environment and continued to grow for 7 d, with surface changes and survival rates recorded. Plants subjected to salt stress were irrigated with 200 mM sodium chloride, and the surface changes in Arabidopsis were observed and recorded, and the plant survival rate was calculated after 7 d. We measured the relevant physiological and biochemical indicators of all lines before and after treatment using the following method to analyze the ability of Arabidopsis to withstand cold and salt stress; the samples used were all fresh samples: We extracted chlorophyll [46] using extraction method and calculated its content [47] according to Lichtenthaler et al. The activity of CAT, SOD, and POD was measured using the method proposed by Ding et al. The enzyme’s activity is expressed in U/g, where U represents unit and catalyzes the generation of 1 unit per minute under optimal or fixed conditions. One unit of enzyme activity is required to produce one mole of product [48]. The MDA content was determined by TBA method [49]. The proline content was measured according to the method used by Qu et al. [50].

### 4.8. Analysis of Gene Expression Related to Stress Resistance

We extracted mRNA from WT, UL, and L1/2/4 lines grown under normal conditions, salt stress, and low-temperature stress, reverse-transcribed them into first-stranded cDNA, used *AtActin* as the internal reference, and detected the functional genes of *Arabidopsis* to MYB through qPCR, reaction process, and reaction system (as shown in Section 4.5). The required primers are recorded in Appendix A.

### 4.9. Statistical Analysis

The data obtained in this study are the average values obtained after three trials, and the differences in the experimental results were obtained through standard deviation, with significance * *p* ≤ 0.05 and ** *p* ≤ 0.01.

## 5. Conclusions

In this study, we obtained a new R2R3-MYB TF gene from *F. vesca* through cloning technology. After analyzing its expression level, we found that the gene has tissue specificity and is more easily expressed in young leaves and roots. After transferring *FvMYB108* into *Arabidopsis*, low-temperature stress and salt stress were applied to the WT, UL, and transgenic lines. Through relevant identification, our hypothesis was confirmed that *FvMYB108* has the ability to regulate plant tolerance to low-temperature and salt stress, and the molecular mechanism by which this gene functions was preliminarily speculated. This discovery provides candidate genes for strawberry molecular breeding, which can be applied to the strawberry industry to expand its planting area, improve the quality of strawberries, and expand the development of the strawberry industry.

## Figures and Tables

**Figure 1 ijms-25-03405-f001:**
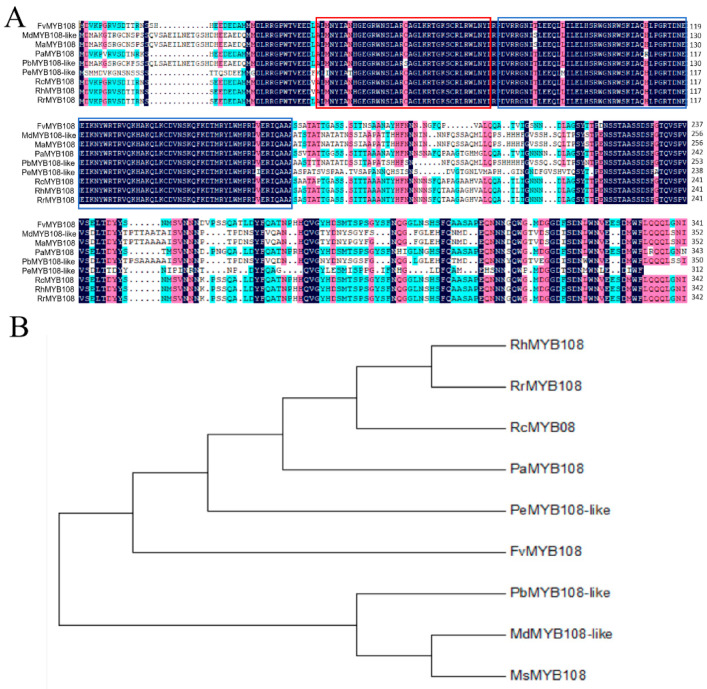
Analysis of the phylogenetic relationship and homologous evolution between FvMYB108 and MYB proteins of other species. (**A**) Analysis of the affinity between FvMYB114 and other MYB proteins. The box indicates two conserved amino acid sequences. (**B**) Evolutionary tree analysis of FvMYB108 and nine other MYB proteins, with the target protein highlighted in red.

**Figure 2 ijms-25-03405-f002:**
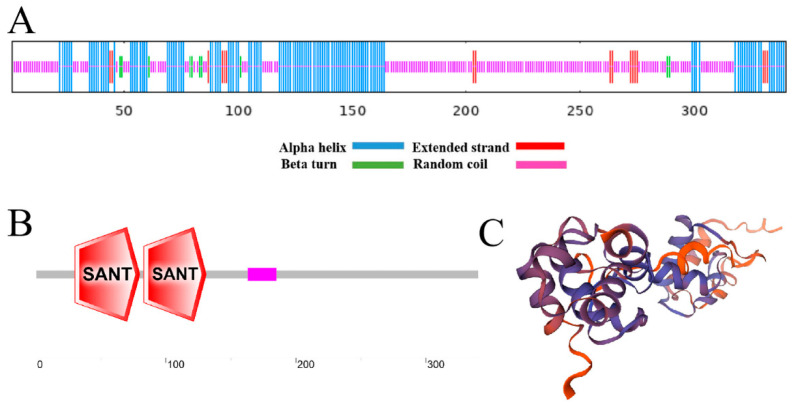
Predicted domain and structure of FVMYB108 protein. (**A**) Secondary structure, (**B**) Domain, and (**C**) Tertiary structure of FVMYB108 protein.

**Figure 3 ijms-25-03405-f003:**
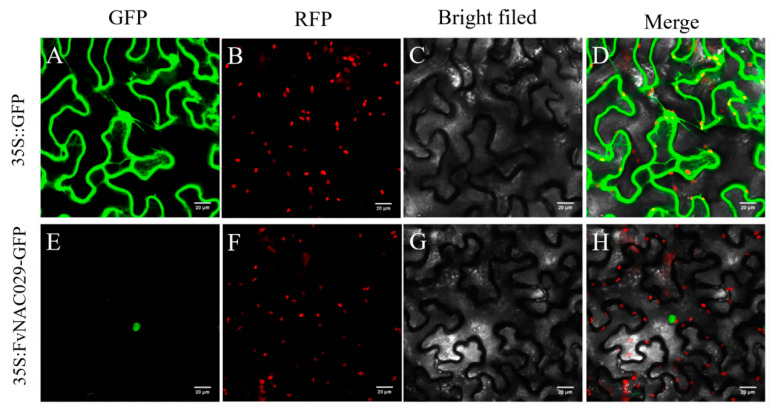
Subcellular localization analysis of FvMYB108. Bar = 20 μM. (**A**,**E**) Fluorescence signals of empty and target genes; (**B**,**F**) The location of the nucleus located in the RFP; (**C**,**G**) The complete morphology of cells in the bright field; (**D**,**H**) The overlap between GFP and RFP.

**Figure 4 ijms-25-03405-f004:**
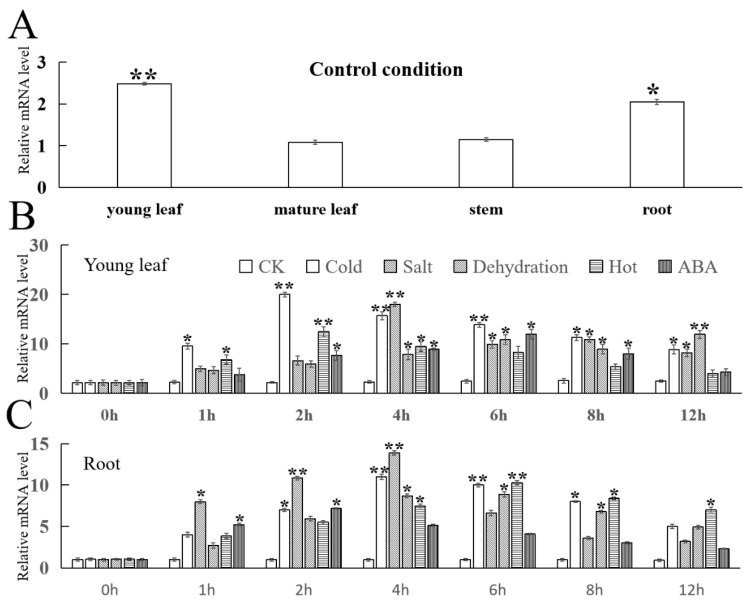
Expression analysis of *FvMYB108* gene. (**A**) Analysis of the expression of *FvMYB108* in different tissues of *F. vesca*. (**B**) Analysis of the expression of *FvMYB108* in young leaves over time under low-temperature stress. (**C**) The expression of *FvMYB108* in roots over time under salt stress. The error line represents the standard deviation * *p* ≤ 0.05, ** *p* ≤ 0.01.

**Figure 5 ijms-25-03405-f005:**
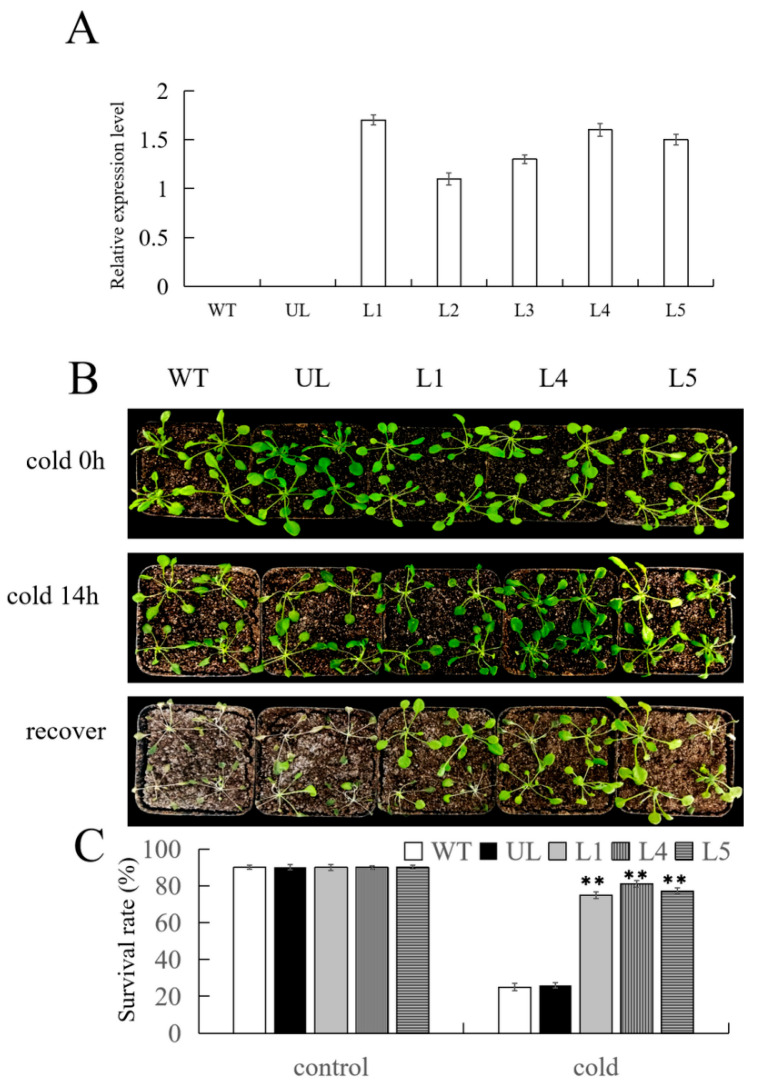
*FvMYB108*-OE *Arabidopsis* has stronger tolerance to cold stress. (**A**) The relative expression levels of *FvMYB108* in WT, UL, and transgenic lines. (**B**) The phenotypes of WT, UL, and transgenic lines before and after low-temperature stress and normal condition recovery. Bar = 5 cm. (**C**) Survival rate of *Arabidopsis* after cold stress, ** *p* ≤ 0.01. Note: WT is a wild-type line; UL line is the vectors transferred in these lines that are not connected to the target gene, and WT and UL were used as controls; L1, L4, and L5 are the transgenic lines used in the experiment.

**Figure 6 ijms-25-03405-f006:**
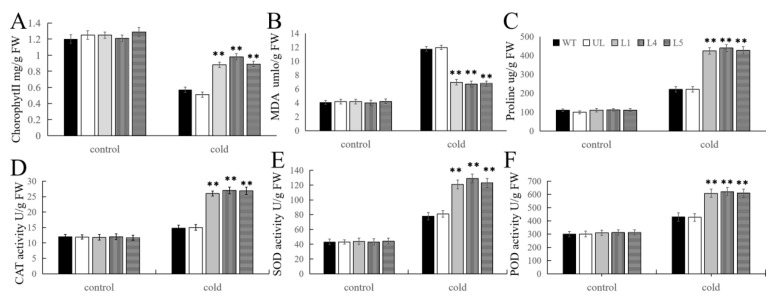
Changes in physiological and biochemical indicators of *FvMYB108*-OE lines before and after low-temperature stress. After growing at −4 °C for 12 h, the content of chlorophyll (**A**), malondialdehyde (**B**), and proline (**C**) changed; the activity changes in (**D**) CAT, (**E**) SOD, and (**F**) POD. ** *p* ≤ 0.01. The control group is the level of each indicator in WT. Note: WT is a wild-type line; UL line is the vectors transferred in these lines that are not connected to the target gene, and WT and UL were used as controls; L1, L4, and L5 are the transgenic lines used in the experiment.

**Figure 7 ijms-25-03405-f007:**
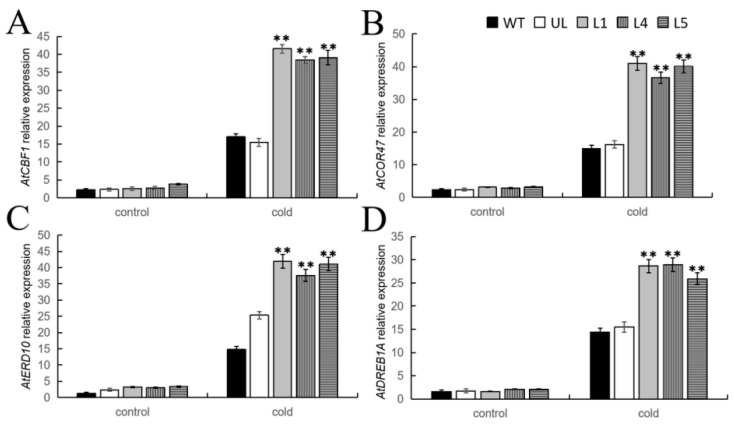
Analysis of the expression levels of genes related to cold stress in various plant lines under low-temperature stress. The relative expression levels of (**A**) *AtCBF1*, (**B**) *AtCOR47*, (**C**) *AtERD10*, and (**D**) *AtDREB1A*. The data are the average of three tests. ** *p* ≤ 0.01. Note: WT is a wild-type line; UL line is the vectors transferred in these lines that are not connected to the target gene, and WT and UL were used as controls; L1, L4, and L5 are the transgenic lines used in the experiment.

**Figure 8 ijms-25-03405-f008:**
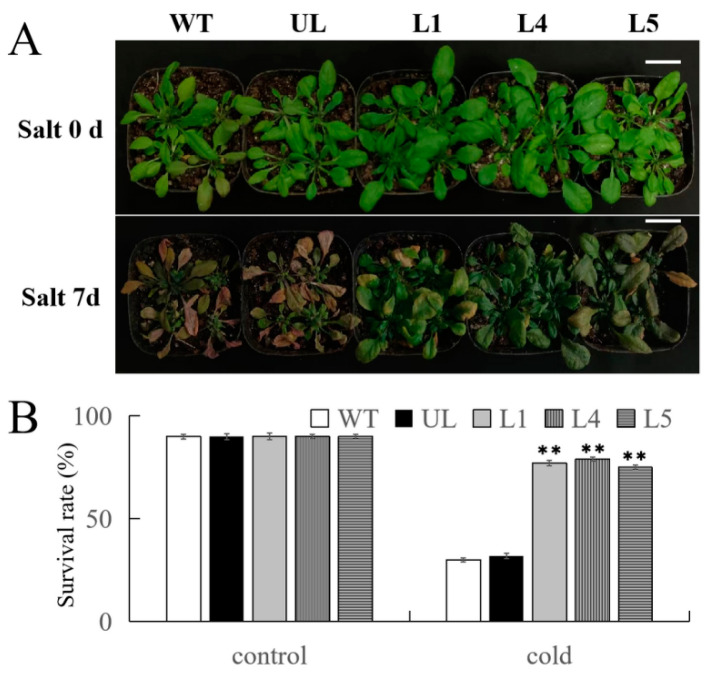
*FvMYB108*-OE *Arabidopsis* has stronger tolerance to salt stress. (**A**) The phenotypes of WT, UL, and transgenic lines before and after salt tress and normal condition recovery. Bar = 5 cm. (**B**) Survival rate of *Arabidopsis* after salt stress, ** *p* ≤ 0.01. Note: WT is a wild-type line; UL line is the vectors transferred in these lines that are not connected to the target gene, and WT and UL were used as controls; L1, L4, and L5 are the transgenic lines used in the experiment.

**Figure 9 ijms-25-03405-f009:**
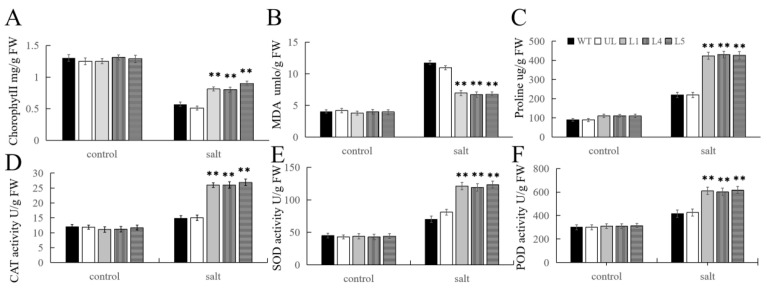
Changes in physiological and biochemical indicators of *FvMYB108*-OE lines before and after low-temperature stress. After 7 d of treatment with 200 mM NaCl, the content of chlorophyll (**A**), malondialdehyde (**B**), and proline (**C**) changed, and the activity changes in (**D**) CAT, (**E**) SOD, and (**F**) POD. ** *p* ≤ 0.01. The control group is the level of each indicator in WT. Note: WT is a wild-type line; UL line is the vectors transferred in these lines that are not connected to the target gene, and WT and UL were used as controls; L1, L4, and L5 are the transgenic lines used in the experiment.

**Figure 10 ijms-25-03405-f010:**
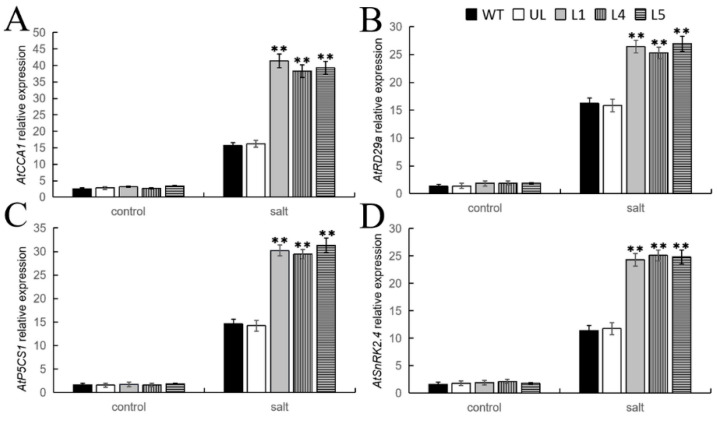
Analysis of the expression levels of genes related to cold stress in various plant lines under salt stress. The relative expression levels of (**A**) *AtCCA1*, (**B**) *AtRD29a*, (**C**) *AtP5CS1*, and (**D**) *AtSnRK2.4*. The data are the average of three tests. ** *p* ≤ 0.01. Note: WT is a wild-type line; UL line is the vectors transferred in these lines that are not connected to the target gene, and WT and UL were used as controls; L1, L4, and L5 are the transgenic lines used in the experiment.

**Figure 11 ijms-25-03405-f011:**
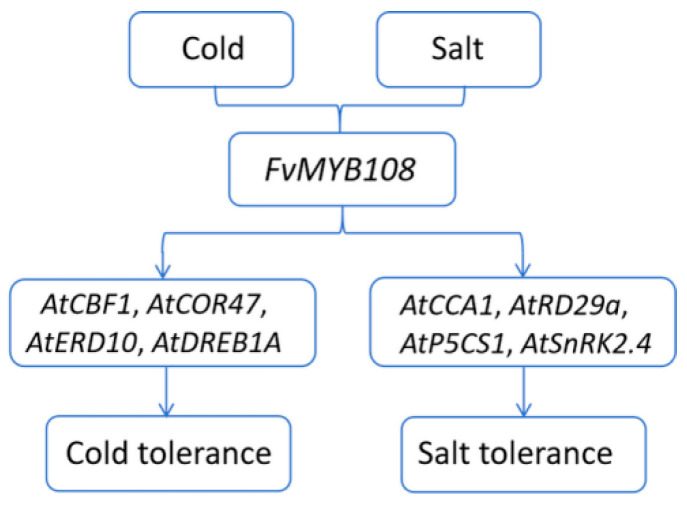
Possible patterns of *FvMYB108* enhancing plant response to cold and salt stress.

## Data Availability

Data are contained within the article and Appendix A.

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
