# Peer review of "FvMYB108, a MYB Gene from Fragaria vesca, Positively Regulates Cold and Salt Tolerance of Arabidopsis"

_ijms, 2024, doi:10.3390/ijms25063405_

Round 1

Reviewer 1 Report

Comments and Suggestions for Authors

The manuscript reported one MYB gene which regulate cold and salt tolerance, and the results are sufficient and the data are reasonable, while the result expression is not scientific, and the language is poor. There are many unscientific and wordy language expressions in the whole manuscript, so the author need to revise the whole manuscript by a native English speaker or an essay polishing agency. And, I have some questions, and found some problems, which need to be paid attention:

1) The title can be revised: A MYB gene from Fragaria vesca, FvMYB108, positively regulates cold and salt tolerance of Arabidopsis.

2) How FvMYB108 was found? Why study this gene? Should be pointed out in the introduction.

3) 2.1. a total length of XXXbp; MdMYB108-like, MsMYB108, PaMYB108, RcMYB08, RhMYB108, RrMYB108, which plant species are these genes from, should be indicated, maybe in M&M section.

4) 2.4. Figure 4A showed the qRT-PCR results. Expressions like this can be deleted.

5) 2.5. what is UL lines? In figure 5A, WT and UL lines have no expression values, why? I think even though WT and UL dont have FvMYB108 gene, qPCR still has expression values. Figure 5C (and Figure 8B), the survival rate of control of the five lines are same? and not 100%?

6) In figure 7, AtSOS1 and AtNHX1, these two genes are related to salt stress, why examine them under cold stress? Also in the model of Figure 11, AtSOS1 and AtNHX1 were shown in cold stress, which is doubtful.

7) Figure 9 D,E,F, only the unit of CAT is U/g min, the others are U/g, why? Whats U? should be clarified in M&M.

8) The author examined gene expressions of AtSOS1, AtCOR47, AtNHX1, AtDREB1A under cold, and AtCCA1, AtRD29a, AtP5CS1, AtSnRK2.4 under salt. The functions of these genes should be clarified, maybe in the discussion, according to previous studies.

9) In the first paragraph of discussion, from ‘ In this study, F. vesca was used’, to ‘Therefore, it can be inferred that the two may have similar functions’, these are all Result, not discussion. Accordingly, in the discussion, the author should rephrased, and improve the depth and breadth of discussion, not only redisplayed the results of this study and other reports.  

Comments on the Quality of English Language

There are many unscientific and wordy language expressions in the whole manuscript, so the author need to revise the whole manuscript by a native English speaker or an essay polishing agency. 

Reviewer 2 Report

Comments and Suggestions for Authors

I have had the opportunity to review the manuscript titled "A MYB gene in Fragaria vesca, FvMYB108, regulates the mechanism of plant cold and salt tolerance" authored by Penghui Song, Ruihua Yang, Kuibao Jiao, Baitao Guo, Lei Zhang, Yuze Li, Kun Zhang, Shuang Zhou, Xinjuan Wu, and Xingguo Li. I find the research presented in this manuscript to be comprehensive and valuable in advancing our understanding of the role of MYB genes in plant responses to abiotic stress, particularly in strawberries.

The study successfully identifies and characterizes a new MYB transcription factor gene, FvMYB108, from Fragaria vesca. The authors conducted thorough bioinformatics analyses, including the determination of its subfamily classification (R2R3-MYB), examination of conserved domains, phylogenetic relationships, predicted protein structure, physicochemical properties, and subcellular localization. These analyses contribute significantly to establishing a comprehensive understanding of the newly discovered FvMYB108 gene.

The experimental design, including the qPCR analysis of FvMYB108 expression in different strawberry organs and subsequent stress treatments, is well-executed. The finding that FvMYB108 is more readily expressed in young leaves and roots, and its increased sensitivity to low temperature and salt stimulation in these organs, is intriguing. The use of transgenic Arabidopsis to assess physiological and biochemical indicators related to stress is a strong approach, and the observed changes provide compelling evidence that FvMYB108 plays a crucial role in enhancing a plant's ability to cope with cold and high salt stress.

 Furthermore, the identification of specific stress-related genes (AtSOS1, AtCOR47, AtNHX1, AtDREB1A, AtCCA1, AtRD29a, AtP5CS1, AtSnRK2.4) that are upregulated upon overexpression of FvMYB108 adds depth to the study. The correlation between FvMYB108 expression and the increased stress tolerance observed in transgenic Arabidopsis plants supports the conclusion that this MYB gene is involved in regulating the plant's response to both low temperature and salt stress.

However, I have a few comments that the authors should take into account:

1) Abstract - please complete this section with a research hypothesis and indicate the practical application of the research obtained.

2) Please list the keywords in alphabetical order. Please do not use words that appear in the title of the manuscript.

3) Introduction - please edit the last paragraph. A research hypothesis should be presented and specific goals outlined.

4) Figure 3 - requires a more detailed explanation below the figure.

5) Figures 5 and 8 - please explain L1, L4 and L5. All figures and tables should be understandable without the need to seek explanations of abbreviations or additional explanations in the text of the manuscript.

6) Figures 6, 7, 9 and 10 - please explain WT, UL, L1, L4 and L5.

7) Figure S1 - it is unnecessarily found in two places in the text of the manuscript.

8) Materials and Methods - Section 4.1. - please use the full scientific name of the species studied.

9) Conclusions - please reword. Please write whether the specific objectives have been met, whether the research hypothesis has been verified, whether the research results have practical application and what is the plan for further research.

10) Please adapt the manuscript more precisely to the template applicable in IJMS, please pay special attention to the References chapter.

11) I noticed that iThenticate has a very high percentage, which I need to lower.

Overall, the manuscript is well-written, and the research is presented in a clear and logical manner. The findings contribute significantly to our understanding of the molecular mechanisms underlying plant cold and salt tolerance. I recommend acceptance of this manuscript for publication in IJMS, as it makes a valuable contribution to the field of plant molecular biology and stress responses.

Round 2

Reviewer 1 Report

Comments and Suggestions for Authors

The author has revised the manuscript following my suggestions. I think it can be pulished now.

Reviewer 2 Report

Comments and Suggestions for Authors

No comments.